# Combination Therapy with a JNK Inhibitor and Hepatocyte Growth Factor for Restoration of Erectile Function in a Rat Model of Cavernosal Nerve Injury: Comparison with a JNK Inhibitor Alone or Hepatocyte Growth Factor Alone

**DOI:** 10.3390/ijms222312698

**Published:** 2021-11-24

**Authors:** Junghoon Lee, Soo Woong Kim, Min Chul Cho

**Affiliations:** 1Department of Urology, Seoul Metropolitan Government Boramae Medical Center, Seoul National University College of Medicine, Seoul 07061, Korea; deftblow@gmail.com; 2Department of Urology, Seoul National University Hospital, Seoul National University College of Medicine, Seoul 03080, Korea; swkim@snu.ac.kr

**Keywords:** erectile dysfunction, apoptosis, endothelium, prostatectomy, hepatocyte growth factor, Jun amino-terminal kinase

## Abstract

We determined if combined administration of JNK-inhibitors and HGF (hepatocyte-growth-factor) would restore erectile-function through both antiapoptotic and regenerative effects in a rat model of cavernous-nerve-crush-injury (CNCI), and compared the results with administration of JNK-inhibitor alone or HGF alone. We randomized 70 rats into 5 groups: sham-surgery-group (S), CNCI (I) group, a group treated with once-daily intraperitoneal-administration of 10.0-mg/kg of JNK-inhibitors (J), a twice-weekly intracavernosal-administration of 4.2-μg HGF group (H), and a combined-treatment with 10.0-mg/kg JNK-inhibitors and 4.2-μg HGF group (J+H). We investigated erectile-responses to electrostimulation, histological-staining, caspase-3-activity-assay, and immunoblotting at two-weeks postoperatively. The three treatment groups showed improvements in erectile-responses (ICP/MAP and AUC/MAP ratios) compared to Group-I. The erectile-responses in Group-J+H were greater than those in Group-J or Group-H. The erectile-responses in Group-J+H were generally normalized. Caspase-3-activity and cJun-phosphorylation in Group-J and Group-J+H improved compared to Group-I, whereas caspase-3-activity in Group-H partially improved. Protein-expression of PECAM-1, eNOS-phosphorylation, and smooth-muscle content in Group-J+H were normalized, although those in Group-J or Group-H were partially restored. Combination therapy with JNK-inhibitors and HGF can generally normalize erectile-function through anti-apoptosis and preservation of endothelium or SM in rat CNCI model. The combined treatment appears to be superior to the respective agent alone in terms of therapeutic effects.

## 1. Introduction

A considerable number of men who undergo radical prostatectomy (RP) due to localized prostate cancer suffers badly from erectile dysfunction following surgery, despite their excellent oncological outcomes [1,2,3]. This post-RP ED, which severely impacts patients’ quality of life, is caused mainly by damages to the cavernous nerves (CNs) during the surgery, although the pathophysiology of post-RP ED is considered to be multifactorial (neurogenic, arteriogenic, venogenic, or a combination thereof) [2]. A previous study reported that the progression of hypogonadism in a post-RP ED rat model could be one of the causes of decreased erectile response [4]. The derangement of penile integrity, including apoptosis of endothelial cells or smooth muscle (SM) cells and progressive fibrosis of the corpus cavernosum, develops in the early period following CN injury [5,6]. These structural derangements of the corpus cavernosum are known to be potential causes of intractable impotence or for poor efficacy of phosphodiesterase type 5 inhibitors (PDE5Is) by inducing cavernosal venoocclusive dysfunction, a key pathophysiology of post-RP ED [2,5].

Almost all therapeutic strategies for penile rehabilitation have failed to prove their efficacy in recent clinical trials, despite their excellent results in preclinical studies [7,8,9]. Studies of stem cell therapies for post-RP ED are still in early clinical stages [10]. It is also necessary to elucidate the long-term safety of stem cells in future clinical trials [10]. According to our previous studies, either Jun N-terminal kinase (JNK) inhibition alone or LIM-kinase 2 (LIMK2) inhibition alone partially restored erectile function in a rat model of CN injury [11,12,13]. In this context, our recent study tested whether the combination of anti-apoptotic effects induced by Jun N-terminal kinase (JNK) inhibition and anti-fibrotic effects of LIM-kinase 2 (LIMK2) inhibition could restore erectile function by rectifying the JNK-driven and LIMK2-driven pathways related to apoptosis and fibrosis in a rat CN crush injury model (CNCI) [14,15]. However, erectile response to electrostimulation or cavernosal venoocclusive function was not restored to normal control levels by the combination therapy in the nerve-injured rats. Recent literature has shown evidence that apoptosis is juxtaposed with fibrosis and contributes importantly to fibrogenesis induced by a variety of stimuli, including transforming growth factor-beta (TGF-β) [16]. Therefore, it would be reasonable to focus on the inhibition of apoptosis rather than fibrosis for the prevention of structural derangements.

The addition of therapeutic strategies for the regeneration of the endothelium and SM to treatments for the suppression of cavernosal apoptosis would be necessary for recovery from CN injury-induced ED [17,18]. HGF (hepatocyte-growth-factor) is a mesenchymal-derived pleiotropic factor that is involved in cell proliferation, differentiation, survival, and angiogenesis [19,20]. Therefore, HGF is a potential substance with regenerative effects on endothelium and SM. In several studies, the vascular remodeling role of HGF improved the cardiovascular system by acting on c-Met receptor, a tyrosine kinase expressed in vascular endothelium and SM [20,21,22]. In addition, our previous study showed that erectile function was partially restored by intracavernosal administration of HGF in a rat CNCI model [17].

We hypothesized that a combined treatment with a JNK inhibitor and HGF would restore erectile function more significantly through both suppression of cavernosal apoptosis and regeneration of cavernosal endothelium or SM in the CN-injured rats, in comparison to a JNK inhibitor treatment alone or HGF alone. The aim of this present study is to determine if the combination of intraperitoneal JNK inhibitor administration and intracavernosal HGF administration, starting in the immediate postinjury period, would restore erectile function through both anti-apoptotic and regenerative effects in an ED rat CNCI model, and to compare this with an intraperitoneal administration of a JNK inhibitor alone or an intracavernosal administration of HGF alone.

## 2. Results

### 2.1. Comparison of Improved Erectile Function Effects with Administration of a JNK Inhibitor Alone, HGF Alone, or Combined Administration of the Two Agents in a Rat CNCI Model

Maximum ICP/MAP and AUC/MAP ratios following stimulation at all voltages in the I group decreased significantly as compared to those in the S group. The three treatment groups (J, H and J+H groups) showed significant improvements in maximum ICP/MAP and AUC/MAP ratios at all stimulation voltages as compared to the I group (Figure 1). There were no differences in the erectile responses following stimulation at all voltages between the J and H groups (Figure 1). Maximum ICP/MAP and AUC/MAP ratios after stimulation at all voltages in the J+H group were greater than those in the J or H groups alone (Figure 1). Except for AUC/MAP following stimulation at 1.0 V, the maximum ICP/MAP and AUC/MAP ratios following stimulation at all voltages in the J+H group did not differ significantly from those in the S group (Figure 1). The maximum ICP/MAP and AUC/MAP ratios following stimulation at all voltages in the J or H group were lower than those in the S group (Figure 1).

### 2.2. Effect of Administration of a JNK Inhibitor Alone, HGF Alone, or Combined Administration of the Two Agents on Structural and Molecular Alterations of the Cavernosal Tissues in a Rat Model of CNCI

Regarding the cavernosal apoptosis, the caspase-3 activity and c-Jun phosphorylation in the I group increased significantly as compared to those in the S group (Figure 1, Figure 2 and Figure 3). The caspase-3 activity and c-Jun phosphorylation in the J and J+H groups were decreased as compared to the I group (Figure 1 and Figure 3). The caspase-3 activity in the J+H group was comparable to that in the S group (Figure 1). The caspase-3 activity in the H group was decreased as compared to that in the I group (Figure 1). There was no difference in c-Jun phosphorylation between the H and I groups (Figure 3).

Regarding endothelial status, PECAM-1 and eNOS phosphorylation protein expression in the I group was significantly decreased as compared to those in the S group (Figure 3). The PECAM-1 and eNOS phosphorylation protein expression in the J and H groups increased significantly as compared to those in the I group but were lower than those in the S group (Figure 3). The cMet phosphorylation in the H and J+H groups was restored to the level observed in the S group (Figure 3). The protein expression of PECAM-1 and eNOS phosphorylation in the J+H group was restored to the levels observed in the S group (Figure 3).

Regarding the SM content, the degree of immunohistochemical staining for α-SMA and SM/collagen ratio in the I group were reduced as compared to those in the S group (Figure 2). The degree of immunohistochemical staining for α-SMA and SM/collagen ratio in the three treatment groups (J, H and J+H groups) were increased as compared to the I group (Figure 2). The degree of immunohistochemical staining for α-SMA in the J and H groups was less than that in the S group, whereas that in the J+H group was restored to the level observed in the S group (Figure 2). The SM/collagen ratios in the three treatment groups (J, H and J+H groups) were increased as compared to those in the I group although they were not restored to the level observed in the S group (Figure 2). The SM/collagen ratio in the J+H group was higher than that in the J or H groups alone (Figure 2).

## 3. Discussion

To date, there have been no definite treatments for post-RP ED or penile rehabilitation in clinical practice. In this context, the present study was designed to suggest a potential treatment strategy for recovery from post-RP ED using a rat model of ED induced by CN injury. We can sum up the results of the present study, as follows: (1) combination therapy with a JNK inhibitor and HGF in the nerve-injured rats could restore erectile function to a level similar to the normal control level. In addition, the degree of improvement in erectile function by the combined administration of the two agents was greater than that by the administration of a JNK inhibitor alone or HGF alone; and (2) the restoration of erectile function by a combination therapy can be mediated by both suppression of cavernosal apoptosis and preservation of endothelium or SM via normalization of JNK-driven and HGF-driven pathways. Thus, the present study extends the current state of knowledge about therapeutic strategies for ED after nerve-sparing RP by suggesting a novel treatment option for ED induced by CN injury. 

Apoptosis in the corpus cavernosum is an important contributor to ED induced by CN injury [5,6]. Previous literature recounts evidence of cavernosal apoptosis immediately following the postinjury period that can last throughout the acute and subacute CN injury period [6,23]. In addition, recent evidence has shown that apoptosis precedes and is required for TGF-β-induced fibrosis [16]. Therefore, suppression of cavernosal apoptosis may be an important part of treatment for recovering from CN injury-induced ED. Although molecular pathways of cavernosal apoptosis induced by CN injuries are still poorly understood, several previous studies, including a previous study of ours, have reported that a few apoptotic pathways including Sonic hedgehog, Rho-kinase, and JNK pathways, are involved in cavernosal apoptosis following CN injury [24,25,26,27]. Under this background, our previous studies reported that JNK pathway inhibition, which was known to play a critical role in apoptosis through both nucleus- targeted and mitochondria-targeted signaling, improved erectile function in a CNCI rat model [11,14,15,28]. However, we thought that both anti-apoptotic and regenerative treatment would be needed to completely preserve the structural integrity of the penis and to recover from CN injury-induced ED, given that the treatment alone for suppressing the structural alterations (apoptosis, fibrosis, or both) of the corpus cavernosum did not normalize erectile function in a CN injury rat model, according to our previous studies [11,14,15]. In accordance with this hypothesis, the present study showed that treatment with a JNK inhibitor alone for two weeks starting in the immediate postinjury period partially restored PECAM-1 protein expression, eNOS phosphorylation, and the erectile response to electrostimulation in a CNCI-induced ED rat model. Meanwhile, HGF, which is reported to have a greater effect on cell regeneration than vascular endothelial growth factor (VEGF) or fibroblast growth factor (FGF), plays an important role in the regeneration of endothelial cells, myocytes, or neurons [20,21,22]. Thus, our recent study showed that the intracavernosal administration of HGF partially restored the erectile responses to electrostimulation in a rat CN injury model [17]. Similarly, a previous study by Das et al. reported that intracavernosal administration of HGF improved erectile function by regeneration of endothelium or neurite through the interaction with cMet in a mouse model of diabetic ED [19]. In line with these studies, this present study showed that treatment with HGF alone partially restored the PECAM-1 protein expression, eNOS phosphorylation, and the erectile function in a CNCI rat model while normalizing cMet phosphorylation. Interestingly, HGF intracavernosal administration appears to have, to some degree, anti-apoptotic effects in the corpus cavernosum, given that it partially improved caspase-3 activity in the corpus cavernosum. Furthermore, the combined treatment with HGF and a JNK inhibitor normalized caspase-3 activity, although the treatment with a JNK inhibitor partially restored it. Taken together, the combined administration of a JNK inhibitor and HGF generally normalized erectile function in a rat CNCI model, according to the present study. The degree of improvement in erectile function by the combination therapy in the nerve-injured rats was also greater than that by the treatment with a JNK inhibitor alone or HGF alone.

The current study is limited by several shortcomings. First, the comparative analyses in the therapeutic effects between the combination therapy and the treatment with a JNK inhibitor alone or HGF alone was performed at only a single time point following CN injury. Subsequent studies in the future will be needed of a well-designed time-series analyses on outcomes of this combination of therapies. Second, there is a difference in the anatomy of the CNs between rats and humans in that the CNs of humans are part of a neurovascular bundle which is difficult to isolate and dissect [29]. Although the rat model is one of the standard research methods, a more ideal animal model may be necessary for subsequent studies about post-RP ED in the future. Third, we did not investigate parameters related to the progression of hypogonadism, such as plasma levels of testosterone and luteinizing hormone, or changes in prostate and testis weight. In the subsequent study, it is needed to evaluate the impact of hypogonadism on ED induced by CN injury. Nevertheless, the current study may have significant clinical implications in that this is the first report on the superiority of a combined treatment with an anti-apoptotic agent and a regenerative agent over an anti-apoptotic agent alone or a regenerative agent alone in terms of restoring erectile function in an animal model of ED induced by CN injury. Thus, this may raise the possibility of combining the two kinds of agents to fully recover from ED (induced by CN injury) through preserving the structural and functional integrity of the penis.

## 4. Materials and Methods

### 4.1. Experimental Animals and Study Design

A total of 70 male Sprague-Dawley rats were randomized into five groups (*n* = 14 in each group), as follows: (S group) rats that underwent sham surgery, (I group) those that underwent CNCI, (J group) those that received once daily intraperitoneal administration of 10.0 mg/kg JNK inhibitors (SP600125, Abcam, Cambridge, UK), (H group) those that received twice weekly intracavernosal administration of 4.2 μg (in PBS 20 μL) recombinant human (rh)-HGF (R&D Systems, Minneapolis, MN, USA), and (J+H group) those that received both once daily intraperitoneal administration of 10.0 mg/kg of JNK inhibitors and twice weekly intracavernosal administration of 4.2 μg (in PBS 20 μL) rh-HGF [11,14,15,17,19]. They were 13 weeks old and weighed 353–411 g at the time of surgery. This current study got approval from the Institutional Animal Care and Use Committee of the Clinical Research Institute at our hospital. We performed all experimental procedures in compliance with the National Research Council guidelines for the care and use of laboratory animals.

In the S group rats, we identified only the CNs running from the major pelvic ganglions (MPGs) on the lateral side of the prostate that had no direct damage after exploring the pelvis. They received daily intraperitoneal administration of saline vehicles and twice-weekly intracavernosal administration of saline vehicles from the day of the surgery. For the I, J, H, and J+H group rats, we crushed the bilateral CNs 4–5 mm below the MPG with mechanical compression (two 80-s applications of pressure), using the microsurgical clamp to mimic nerve-sparing RP, as previously described [14,15]. The I group rats received daily intraperitoneal administration of saline vehicles and twice-weekly intracavernosal administration of saline vehicles starting on the day of the CNCI. The J group rats received daily intraperitoneal administration of 10 mg/kg JNK inhibitors and twice-weekly intracavernosal administration of saline vehicles starting on the day of the CNCI. The H group rats received daily intraperitoneal administration of saline vehicles and twice-weekly intracavernosal administration of 4.2 μg (in PBS 20 μL) rh-HGF starting on the day of the CNCI. The J+H group rats received daily intraperitoneal administration of 10 mg/kg JNK inhibitors and twice-weekly intracavernosal administration of 4.2 μg (in PBS 20 μL) rh-HGF starting on the day of the CNCI. The drug administration was interrupted for a 48-h washout period prior to the in vivo assessment of erectile function at two weeks following surgery.

### 4.2. Assessment of In Vivo Erectile Function and the Collection of Penile Tissues

We assessed erectile function in seven animals from each group using erectile responses to electrostimulation in a routine manner [14]. Briefly, after the animals were anesthetized with 10 mg/kg Zoletil (Virbac Laboratories, Carros, France), the right carotid artery and the corpus cavernosum were cannulated using a polyethylene (PE)-50 tube filled with heparinized saline for monitoring mean arterial pressure (MAP) and intracavernosal pressure (ICP). For continuously measuring the ICP and MAP, the PE-50 tubes were connected to the pressure transducer attached to the high-performance data acquisition hardware (PowerLab; ADInstruments, Dunedin, New Zealand). Erectile responses were obtained by electrostimulation of the CNs distal to the point of CNCI. The electrostimulation parameters were 1.0 V, 3.0 V, or 5.0 V, with 15 Hz and a pulse width of 0.2 ms for 30 s. Since the ICP and area under the curve (AUC) during the entire erectile response were normalized by the MAP, the erectile responses were expressed as maximum ICP/MAP and AUC/MAP.

In order to avoid inadvertent changes in the caspase-3 activity assay, histological staining and immunoblotting studies, total penectomy was performed in the remaining seven animals from each group in which in vivo erectile function tests were not performed. The middle parts of the skin-denuded undamaged penile shaft were collected, fixed in 10% formaldehyde solution (Sigma-Aldrich, St. Louis, MO, USA), and embedded in paraffin wax (Sigma-Aldrich) for histological staining. The remaining penile tissues were preserved at −80 °C until processing for subsequent molecular studies.

#### Determination of Caspase-3 Activity

For comparing the degree of cavernosal apoptosis among the groups, the activity of caspase-3 (an executioner enzyme of apoptosis) in the corpus cavernosum was quantitatively measured in seven animals from each group using a caspase-3 colorimetric assay kit (Abcam) in a routine manner [15]. The data was presented as being many times greater than the controls.

### 4.3. Evaluation of Structural Alterations in Corpus Cavernosum

We did immunohistochemical staining for α-SM actin (α-SMA) and the Masson’s trichrome staining in a routine manner, in order to determine the SM content and the SM/collagen ratio [14,15] After paraffin-embedded penile tissue sections had been deparaffinized, they were stained with a primary antibody against α-SMA (1:100, Dako, Glostrup, Denmark) or Masson’s trichrome. In the immunohistochemical staining for α-SMA, the proportions of the SM area (stained in brown) in a given area comprising half of the corpus cavernosum were evaluated at 40× magnification using Image Pro Plus 4.5 software (Medica Cybernetics, Silver Spring, MD, USA). In the Masson’s trichrome staining, the ratios of the SM area (stained in red)/the collagen area (stained in blue) was determined at 40× magnification of the penile tissue (half the corpus cavernosum). An independent observer conducted the quantitative image analyses for the SM content and the SM/collagen ratio at each stained slide in a blind fashion. 

### 4.4. Immunoblot Analysis

Immunoblotting was performed to quantitatively measure the protein expression of molecules (c-Jun, PECAM-1, eNOS and cMet) related to the JNK, endothelium, and HGF. Primary antibodies used in this study were anti-c-Jun (1:1000, Cell-Signaling Technology, Danvers, MA, USA), anti-phospho-c-Jun (1:1000, Cell-Signaling Technology), anti-PECAM-1 (1:1000 dilution, Cell-Signaling Technology), anti-total-eNOS (1:2000 dilution, Cell Signaling), anti-phospho-eNOS (1:1000 dilution, Cell Signaling), anti-total-cMet (1:2000 dilution, Abcam), and anti-phospho-cMet (1:1000 dilution, Abcam). The relative gray levels of the molecules to β-actin were quantitatively measured by densitometric analysis using ImageJ software (U.S. National Institutes of Health, Bethesda, MD, USA).

### 4.5. Data Analyses

All data represented means ± standard errors of the mean. We tested the difference between the experimental groups using the Kruskal-Wallis test and Mann-Whitney test for post-hoc analysis. We used Bonferroni correction for adjusting multiple comparisons. Statistical significance was set at a level of p values less than 0.05. SPSS Version 20.0 (SPSS Inc., Chicago, IL, USA) was utilized for data analysis.

## 5. Conclusions

The present study indicates that combination therapy with a JNK inhibitor and HGF can normalize erectile function in a CN injury rat model. This therapeutic effect appears to be mediated by anti-apoptosis and preservation of endothelium or SM in the corpus cavernosum via normalization of JNK-driven and HGF-driven pathways. Moreover, the combined treatment with the two kinds of agents appears to be superior to the respective agents alone in terms of their therapeutic effects. Further well-designed time-series studies with longer follow-up are needed to validate our results.

## Figures and Tables

**Figure 1 ijms-22-12698-f001:**
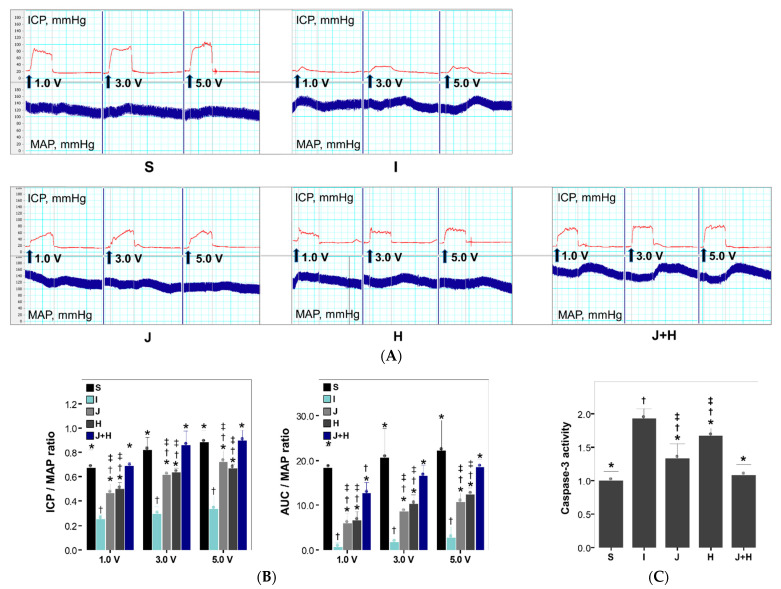
Comparison of erectile function and the caspase-3 activity among the five experimental groups at two weeks after surgery. (**A**) Representative ICP responses to electrostimulation in each experimental group. (**B**) Bar graphs presenting a comparison in erectile responses to electrostimulation (maximum ICP/MAP and AUC/MAP) among the five experimental groups. The data were presented as the mean ± standard errors of the mean (SEM). (**C**) A bar graph showing comparative analyses in the caspase-3 activity. S: a group in which sham surgery was performed, I: a group in which the cavernous nerve crush injury was performed, J: a group in which a 10.0 mg/kg JNK inhibitor was administered daily from the day of the cavernous nerve crush injury, H: a group in which 4.2 μg hepatocyte growth factor was administered twice-weekly from the day of the cavernous nerve crush injury, J+H: a group in which a 10.0 mg/kg JNK inhibitor and 4.2 μg hepatocyte growth factor were administered daily and twice-weekly from the day of the cavernous nerve crush injury, respectively. ICP, intracavernosal pressure; MAP, mean arterial pressure; AUC, area under the curve. * *p* < 0.05: compared to the I group, † *p* < 0.05: compared to the S group, ‡ *p* < 0.05: compared to the J+H group.

**Figure 2 ijms-22-12698-f002:**
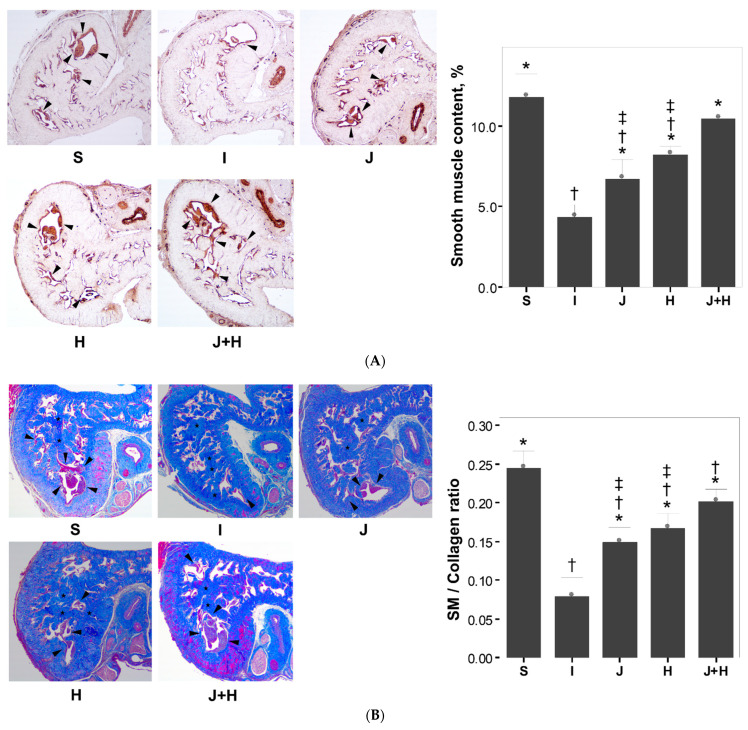
Comparison of the smooth muscle content and the smooth muscle/collagen ratio at two weeks after surgery among the five experimental groups. Bar graphs showing comparative analyses in the degree of immunohistochemical staining for α-SMA (**A**) and the smooth muscle/collagen ratio among the experimental groups (**B**). The data were presented as the mean ± standard errors of the mean (SEM). S, a group in which sham surgery was performed; I, a group in which the cavernous nerve crush injury was performed; J, a group in which a 10.0 mg/kg JNK inhibitor was administered daily from the day of the cavernous nerve crush injury; H, a group in which 4.2 μg hepatocyte growth factor was administered twice-weekly from the day of the cavernous nerve crush injury, J+H: a group in which a 10.0 mg/kg JNK inhibitor and 4.2 μg hepatocyte growth factor were administered daily and twice-weekly from the day of the cavernous nerve crush injury, respectively. α-SMA, alpha-smooth muscle actin; SM, smooth muscle. * *p* < 0.05: compared to the I group, † *p* < 0.05: compared to the S group, ‡ *p* < 0.05: compared to the J+H group.

**Figure 3 ijms-22-12698-f003:**
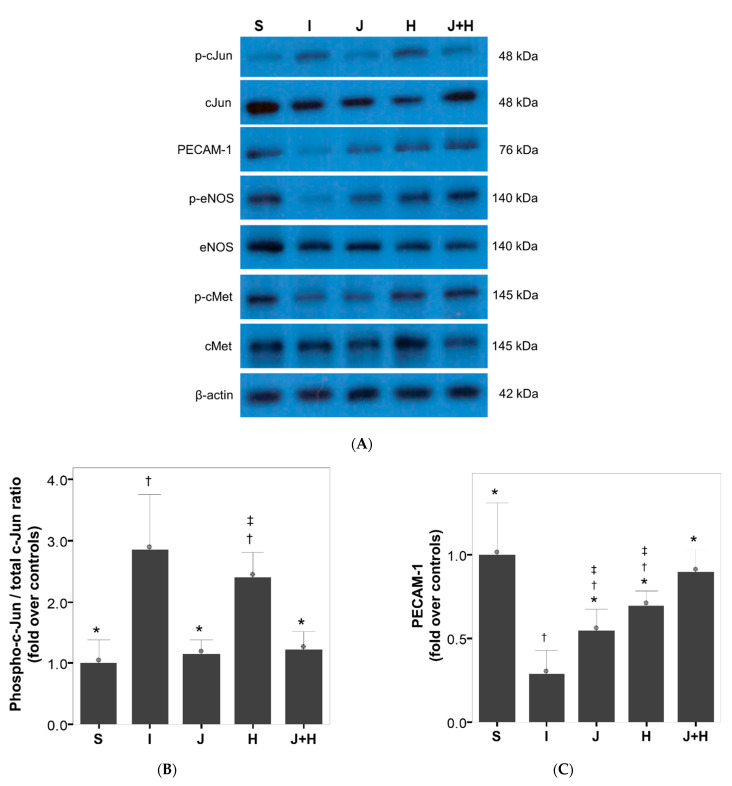
Comparison in the protein expression of molecules (cJun, PECAM-1, eNOS, and cMet) related to the JNK, endothelium and HGF among the five experimental groups. Representative Western blots shows the expression level of the protein molecules (**A**). Bar graphs presenting comparative analyses in cJun phosphorylation (**B**), the protein expression of PECAM-1 (**C**), eNOS phosphorylation (**D**), and cMet phosphorylation among the experimental groups (**E**). The data were presented as the mean ± standard errors of the mean (SEM). The values were normalized to those of β-actin and presented as fold changes over controls. S, a group in which sham surgery was performed; I, a group in which the cavernous nerve crush injury was performed; J, a group in which a 10.0 mg/kg JNK inhibitor was administered daily from the day of the cavernous nerve crush injury; H, a group in which 4.2 μg hepatocyte growth factor was administered twice-weekly from the day of the cavernous nerve crush injury; J+H, a group in which a 10.0 mg/kg JNK inhibitor and 4.2 μg hepatocyte growth factor were administered daily and twice-weekly from the day of the cavernous nerve crush injury, respectively. JNK, Jun N-terminal kinase; eNOS, endothelial nitric oxide synthase. * *p* < 0.05: compared to the I group, † *p* < 0.05: compared to the S group, ‡ *p* < 0.05: compared to the J+H group.

## Data Availability

The datasets generated during and/or analyzed during the current study are available from the corresponding author on reasonable request.

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
