# Peer review of "Combination Therapy with a JNK Inhibitor and Hepatocyte Growth Factor for Restoration of Erectile Function in a Rat Model of Cavernosal Nerve Injury: Comparison with a JNK Inhibitor Alone or Hepatocyte Growth Factor Alone"

_ijms, 2021, doi:10.3390/ijms222312698_

Round 1

Reviewer 1 Report

Considering the emotional and mental health issues and the physical morbidity associated with post-RP erectile dysfunction, a common postoperative complication of RP, the current study is a worthwhile effort. This a reasonably well-constructed study with quantitative analysis of several important pathophysiological changes associated with the post-RP ED. Several points suggested here will improve the quality of the manuscript:

1) Authors recently published that a combination of JNK inhibitors and LIMK2 blockers may partially suppress cavernous apoptosis and fibrosis and therefore, may help with restoring ED. It is not clear, why they choose to study HGF along with JNK inhibitors in the current study. Was LIMK2 blockade ineffective in the previous study? A paper indicating beneficial effects of HGF administration has been cited. A clear rationale for employing JNK blockers and HGF should be stated. In line 62, should introduce HGF and explain the reason for using HGF in the current study.

In lines 76 and 77, please explain reasons for employing doses and frequency of administration of JNK inhibitor and HGF in the current study. Why a dose-response study was not considered?

In lines 91 and 92, explain how this model is comparable to human post-RP ED which likely involve unintentional resection of cavernous nerve. A crush injury as used in the current study be similar to transection of nerves?

Figure 3A: lanes should be labelled.

Line 248, please qualify the statement that ...post-RP ED where cavernous nerves are not severed or transected.

In line 293, authors have identified that" only a single time point was used". What was the rationale of choosing this time point? 

Author Response

Reviewer 1

Authors recently published that a combination of JNK inhibitors and LIMK2 blockers may partially suppress cavernous apoptosis and fibrosis and therefore, may help with restoring ED.

1) It is not clear, why they choose to study HGF along with JNK inhibitors in the current study. Was LIMK2 blockade ineffective in the previous study?

  • We appreciate the reviewer’s good comment. In our previous study, when 10 mg/kg JNK inhibitor was administered for 2 weeks, ICP/MAP and AUC/MAP were improved in all voltage stimulations, but they were not restored to the control level (Figure 1) [1]. Also, in a study in which 10 mg/kg LIMK2 inhibitor was administered for 1 week, ICP/MAP and AUC/MAP were improved in all voltage stimulations, but they were not restored to the control level (Figure 2) [2]. In addition, after chronic treatment with 10 mg/kg LIMK2 inhibitor for 30 days, an improvement in erectile function was observed in papaverine response, maintenance rate, and drop rate of dynamic-infusion-cavernosometry (Figure 3) [3]. However, chronic inhibition of LIMK2 did not completely restore dynamic-infusion-cavernosometry parameters to normal control levels. We additionally cite and describe these studies in line 51 of the Introduction section as follow: “According to our previous studies, Jun N-terminal kinase (JNK) inhibition alone or LIM-kinase 2 (LIMK2) inhibition alone partially restored erectile function in a rat model of CN injury [1,2,3]”

Figure 1. The comparison in erectile responses to electrostimulation at 2 weeks after surgery. L: CNCI treated with 1 mg/kg JNK inhibitor, H: with 10 mg/kg JNK inhibitor

- Please see the submitted pdf file.

Figure 2. The comparison in erectile responses to electrostimulation at 1 weeks after surgery. L: CNCI treated with 2.5 mg/kg LIMK2 inhibitor, M: with 5 mg/kg LIMK2 inhibitor, H: with 10 mg/kg LIMK2 inhibitor

- Please see the submitted pdf file.

Figure 3. The comparison in dynamic-infusion-cavernosometry at 30 days after surgery. (a) Papaverine response, (b) Maintenance rate, (c) Drop rate. L: CNCI treated with 10 mg/kg LIMK2 inhibitor

- Please see the submitted pdf file.

  1. Park J, Chai JS, Kim SW, Paick JS, Cho MC. Inhibition of Jun N-terminal Kinase Improves Erectile Function by Alleviation of Cavernosal Apoptosis in a Rat Model of Cavernous Nerve Injury. Urology 2018;113:253.e9-253.e16.
  2. Park J, Cho SY, Park K, et al. Role of inhibiting LIM-kinase2 in improving erectile function through suppression of corporal fibrosis in a rat model of cavernous nerve injury. Asian J Androl. 2018 Jul-Aug: 20:372-8
  3. Park J, Son H, Chai JS, Kim SW, Paick JS, Cho MC. Chronic administration of LIMK2 inhibitors alleviates cavernosal veno-occlusive dysfunction through suppression of cavernosal fibrosis in a rat model of erectile dysfunction after cavernosal nerve injury. PLoS One. 2019: 14:e0213586

2) A paper indicating beneficial effects of HGF administration has been cited. A clear rationale for employing JNK blockers and HGF should be stated. In line 62, should introduce HGF and explain the reason for using HGF in the current study.

  • We agree with the reviewer’s comment. HGF is a potential substance with regenerative effects on endothelium and smooth muscle. In a previous study, we reported that a rat model of CNCI administered with HGF alone showed partial recovery of erectile function in a dose-dependent manner. As the reviewer suggested, we revised the manuscript of Introduction section, as follow in line 68:

“HGF (hepatocyte-growth-factor) is a mesenchymal-derived pleiotropic factor that is involved in cell proliferation, differentiation, survival, and angiogenesis [1, 2]. Therefore, HGF is a potential substance with regenerative effects on endothelium and SM. In several studies, the vascular remodeling role of HGF improved the cardiovascular system by acting on c-Met receptor, a tyrosine kinase expressed in vascular endothelium and SM [2,3, 4]. In addition, our previous study showed that erectile function was partially restored by intracavernosal administration of HGF in a rat CNCI model [5].”

  • We appreciate the reviewer’s good comment.
  1. Das ND, Yin GN, Choi MJ, Song KM, Park JM, Limanjaya A, Ghatak K, Minh NN, Ock J, Park SH, Kim HM, Ryu JK, Suh JK. Effectiveness of Intracavernous Delivery of Recombinant Human Hepatocyte Growth Factor on Erectile Function in the Streptozotocin-Induced Diabetic Mouse. J Sex Med 2016;13(11):1618-1628.
  2. Nakamura Y, Morishita R, Higaki J, Kida I, Aoki M, Moriguchi A, Yamada K, Hayashi S, Yo Y, Nakano H, Matsumoto K, Nakamura T, Ogihara T. Hepatocyte growth factor is a novel member of the endothelium-specific growth factors: additive stimulatory effect of hepatocyte growth factor with basic fibroblast growth factor but not with vascular endothelial growth factor. J Hypertens 1996;14(9):1067-72.
  3. Gallo S, Sala V, Gatti S, Crepaldi T. Cellular and molecular mechanisms of HGF/Met in the cardiovascular system. Clin Sci (Lond) 2015;129(12):1173-93.
  4. Sakai K, Aoki S, Matsumoto K. Hepatocyte growth factor and Met in drug discovery. J Biochem 2015;157(5):271-84.
  5. Yoo S, Park J, Son H, Kim SW, Paick JS, Cho MC. Improvement of erectile function by intracavernous injection of hepato-cyte growth factor in a rat model of erectile dysfunction after cavernous nerve injury. J Sex Med 2018;15(7):S319. doi: 10.1016/j.jsxm.2018.04.437

3) lines 76 and 77, please explain reasons for employing doses and frequency of administration of JNK inhibitor and HGF in the current study. Why a dose-response study was not considered?

  • We agree with the reviewer’s comment. We referenced the dose and frequency of JNK inhibitor and HGF that we and other researchers used in previous studies to show good effect [1, 2, 3, 4, 5]. As the reviewer suggested, we added more previous studies as references in line 101 of the Materials and Methods section. The dose and frequency of administration of the JNK inhibitor were based on our previous studies, and 10 mg/kg of the JNK inhibitor was administered intraperitoneal once a day immediately after surgery. The dose of HGF was determined to be 4.2 μg by referring to the study results of Das et al. [3]. They set the frequency to be administered twice before surgery (day -3 and day 0 of 2 weeks), but we increased the frequency to four times (twice per week for 2 weeks) considering the short half-life of HGF.
  • We appreciate the reviewer’s good comment.
  1. Kim SW, Lee J, Park J, Chai JS, Oh S, Paick JS, Cho MC. Combination of LIM-kinase 2 and Jun Amino-terminal Kinase Inhibitors Improves Erectile Function in a Rat Model of Cavernous Nerve Injury. Urology 2019;131:136-143.
  2. Cho MC, Lee J, Park J, Kim SW. Restoration of Cavernous Veno-Occlusive Function through Chronic Administration of a Jun-Amino Terminal Kinase Inhibitor and a LIM-Kinase 2 Inhibitor by Suppressing Cavernous Apoptosis and Fibrosis in a Rat Model of Cavernous Nerve Injury: A Comparison with a Phosphodiesterase Type 5 Inhibitor. World J Mens Health. 2020 Jul 9. doi: 10.5534/wjmh.200085. Online ahead of print.
  3. Das ND, Yin GN, Choi MJ, Song KM, Park JM, Limanjaya A, Ghatak K, Minh NN, Ock J, Park SH, Kim HM, Ryu JK, Suh JK. Effectiveness of Intracavernous Delivery of Recombinant Human Hepatocyte Growth Factor on Erectile Function in the Streptozotocin-Induced Diabetic Mouse. J Sex Med 2016;13(11):1618-1628.
  4. Park J, Chai JS, Kim SW, Paick JS, Cho MC. Inhibition of Jun N-terminal Kinase Improves Erectile Function by Alleviation of Cavernosal Apoptosis in a Rat Model of Cavernous Nerve Injury. Urology 2018;113:253.e9-253.e16.
  5. Yoo S, Park J, Son H, Kim SW, Paick JS, Cho MC. Improvement of erectile function by intracavernous injection of hepatocyte growth factor in a rat model of erectile dysfunction after cavernous nerve injury. J Sex Med 2018;15(7):S319. doi: 10.1016/j.jsxm.2018.04.437

4) In lines 91 and 92, explain how this model is comparable to human post-RP ED which likely involve unintentional resection of cavernous nerve. A crush injury as used in the current study be similar to transection of nerves?

  • We appreciate the reviewer’s good comment. In the rat model, crush injury (CN), cautery, freeze injury, transection, and excision have been used to mimic possible damage that occurs during the radical prostatectomy procedure [1]. Both crush and freeze CN injury model appeared to replicate the type of CN injury seen following nerve-sparing RP [2]. CN transection model replicates that of non-nerve-sparing RP [2]. And in the meantime, many studies have used CN crush injury as a method to mimic nerve-sparing RP [3,4], and CN transection injury has been used as a method to mimic non-nerve-sparing RP [5. 6].
  1. Haney NM, Nguyen HMT, Honda M, Abdel-Mageed AB, Hellstrom WJG. Bilateral Cavernous Nerve Crush Injury in the Rat Model: A Comparative Review of Pharmacologic Interventions. Sex Med Rev. 2018 Apr: 6:234-41
  2. Chung E, De Young L, Brock GB. Investigative models in erectile dysfunction: a state-of-the-art review of current animal models. J Sex Med. 2011 Dec: 8:3291-305
  3. Jin HR, Chung YG, Kim WJ, et al. A mouse model of cavernous nerve injury-induced erectile dysfunction: functional and morphological characterization of the corpus cavernosum. J Sex Med. 2010 Oct: 7:3351-64
  4. Hsieh PS, Bochinski DJ, Lin GT, Nunes L, Lin CS, Lue TF. The effect of vascular endothelial growth factor and brain-derived neurotrophic factor on cavernosal nerve regeneration in a nerve-crush rat model. BJU Int. 2003 Sep: 92:470-5
  5. Lysiak JJ, Yang SK, Klausner AP, Son H, Tuttle JB, Steers WD. Tadalafil increases Akt and extracellular signal-regulated kinase 1/2 activation, and prevents apoptotic cell death in the penis following denervation. J Urol. 2008 Feb: 179:779-85
  6. Mullerad M, Donohue JF, Li PS, Scardino PT, Mulhall JP. Functional sequelae of cavernous nerve injury in the rat: is there model dependency. J Sex Med. 2006 Jan: 3:77-83

5) Figure 3A: lanes should be labelled.

  • We appreciate the reviewer’s good comment. We revised it. Please see the submitted pdf file.

6) Line 248, please qualify the statement that ...post-RP ED where cavernous nerves are not severed or transected.

  • We agree with the reviewer’s comment. We revised “post-RP ED” to “ED after nerve-sparing RP”. We appreciate the reviewer’s good comment.

7) In line 293, authors have identified that" only a single time point was used". What was the rationale of choosing this time point?

  • We appreciate the reviewer’s good comment. User et al. reported that apoptosis is most active on the 2nd day after injury, mainly occurs within 1-2 weeks, and then stabilizes [1]. These results were also invesigated in our previously published study [2]. Based on the results of these studies, previous studies related to the evaluation of apoptosis were evaluated 2 weeks after injury. And we especially referred to the results of previous studies conducted at 2 weeks with JNK inhibitors or HGF [3, 4]. Das et al also evaluated 2 weeks after HGF administration [5]. Therefore, we decided to evaluate this study at 2 weeks after injury for consistency and comparison with other studies.
  1. User HM, Hairston JH, Zelner DJ, McKenna KE, McVary KT. Penile weight and cell subtype specific changes in a post-radical prostatectomy model of erectile dysfunction. J Urol. 2003 Mar: 169:1175-9
  2. Kim TB, Cho MC, Paick J-S, Kim SW. Is It Possible to Recover Erectile Function Spontaneously after Cavernous Nerve Injury? Time-Dependent Structural and Functional Changes in Corpus Cavernosum Following Cavernous Nerve Injury in Rats. Korean J Androl. 2012 4/: 30:31-9
  3. Park J, Chai JS, Kim SW, Paick JS, Cho MC. Inhibition of Jun N-terminal Kinase Improves Erectile Function by Alleviation of Cavernosal Apoptosis in a Rat Model of Cavernous Nerve Injury. Urology. 2018 Mar: 113:253 e9- e16
  4. Yoo S, Park J, Son H, Kim SW, Paick JS, Cho MC. Improvement of erectile function by intracavernous injection of hepato-cyte growth factor in a rat model of erectile dysfunction after cavernous nerve injury. J Sex Med 2018;15(7):S319. doi: 10.1016/j.jsxm.2018.04.437
  5. Das ND, Yin GN, Choi MJ, et al. Effectiveness of Intracavernous Delivery of Recombinant Human Hepatocyte Growth Factor on Erectile Function in the Streptozotocin-Induced Diabetic Mouse. J Sex Med. 2016 Nov: 13:1618-28

We thank you for the invaluable comments and helpful suggestions that contributed to revision of our manuscript.

Reviewer 2 Report

In this manuscript it is described the effects of the combination of intraperitoneal JNK inhibitor administration and intracavernosal HGF administration on erectile function in a rat model of cavernous-nerve-crush-injury (CNCI). Howewer the authors compared these effects with administration of JNK-inhibitor alone or HGF alone. This manuscript contains some interesting observations and it is the opinion of this reviewer that it is could be suitable for publication provided that the Authors are willing to revise the following points. In fact, although the topic is interesting, results reported are not sound enough to add compelling information to the reader.

There are indeed some problems:

  • Hypogonadism is often associated with CNCI. In a rat model of bilateral cavernous neurotomy (BCN), Vignozzi et al. observed the onset of an overt condition of hypogonadism, characterized by reduced T plasma level, reduced ventral prostate weight, reduced testis function (including testis weight and number of Leydig cells), with an inadequate compensatory increase of luteinizing hormone (see Vignozzi et al., Sex Med 2009 May;6(5):1270-83. Please add a table with testosterone, LH plasma levels and the weight of androgen target tissues, as seminal vesicles and prostate in all groups. These results can help to discuss the results obtained with drugs used.
  • The measurement of erectile response elicited by electrical stimulation (ES) of cavernous nerve was performed at 15 Hz. Which is the effect at varying frequencies (1, 2, 4, 8, 16, 32Hz) and in particular at low frequencies?

Minor points:

The language will require correction throughout.

Author Response

Reviewer 2

Hypogonadism is often associated with CNCI. In a rat model of bilateral cavernous neurotomy (BCN), Vignozzi et al. observed the onset of an overt condition of hypogonadism, characterized by reduced T plasma level, reduced ventral prostate weight, reduced testis function (including testis weight and number of Leydig cells), with an inadequate compensatory increase of luteinizing hormone (see Vignozzi et al., Sex Med 2009 May;6(5):1270-83.

1) Please add a table with testosterone, LH plasma levels and the weight of androgen target tissues, as seminal vesicles and prostate in all groups. These results can help to discuss the results obtained with drugs used.

  • We agree with the reviewer’s comment. The reviewer’s comment that hypogonadism that progresses after post-RP may be the cause of ED is an important consideration in the rat model of post-RP ED. Unfortunately, it is difficult to obtain additional data about plasma levels of testosterone and luteinizing hormone, or changes in prostate and testis weight because all of the experimental animals were already sacrificed. Therefore, In the introduction section, we introduce and cite Vignozzi's study that the progression of hypogonadism after cavernous nerve injury may be one cause of post-RP ED in line 39. We also added in the Limitations in the Discussion section of the revised manuscript. Because the reviewer’s comment is very important, we will evaluate the factors related to the progression of hypogonadism to the study design in the future studies. We appreciate the reviewer’s good comment.

Line 39: “A previous study reported that the progression of hypogonadism in a post-RP ED rat model could may be one of the causes of decreased erectile response [1].”

Line 313: "Third, we did not investigate parameters related to the progression of hypogonadism, such as plasma levels of testosterone and luteinizing hormone, or changes in prostate and testis weight. In the subsequent study, it is needed to evaluate the impact of hypogonadism on ED induced by CN injury."

1. Vignozzi L, Filippi S, Morelli A, et al. Cavernous neurotomy in the rat is associated with the onset of an overt condition of hypogonadism. J Sex Med. 2009 May: 6:1270-83

2) The measurement of erectile response elicited by electrical stimulation (ES) of cavernous nerve was performed at 15 Hz. Which is the effect at varying frequencies (1, 2, 4, 8, 16, 32Hz) and in particular at low frequencies?

  • We appreciate the reviewer’s good comment. The electrical stimulation parameters used by different research groups show high variability. The European Society for Sexual Medicine Consensus Statement descripted that the most commonly agreed parameter ranges for cavernous nerve stimulation in rats are frequency, 10-20Hz; voltage, 2.5-8.0V [1]. Ueno et al. reported that (Mean Intracavernosal Pressure/Mean Arterial Pressure) gradually increased from 1 Hz to 12 Hz, but decreased from 12 Hz to 30 Hz (Figure 1) [2]. They also stated that the reason that the effect of sildenafil is evident at lower frequencies is that the cavernous smooth muscles and penile deep arteries are already maximally relaxed by strong electrical stimulation at higher frequencies. Furthermore, our previous experiments showed that the erectile responses by electrical stimulation were voltage-dependent. Therefore, in the present study, the frequency of electrostimulation was set to 15Hz, and the erectile responses were investigated at varying voltages (1.0, 3.0 and 5.0 voltages).

Figure1. Please see the submitted pdf file.

  1. Weyne E, Ilg MM, Cakir OO, et al. European Society for Sexual Medicine Consensus Statement on the Use of the Cavernous Nerve Injury Rodent Model to Study Postradical Prostatectomy Erectile Dysfunction. Sex Med. 2020 Sep: 8:327-37
  2. Ueno N, Iwamoto Y, Segawa N, Kinoshita M, Ueda H, Katsuoka Y. The effect of sildenafil on electrostimulation-induced erection in the rat model. Int J Impot Res. 2002 Aug: 14:251-5

Minor points:

3) The language will require correction throughout.

  • We appreciate the reviewer’s comment. In order to meet the deadline of revision requested by the journal, we submit the revised manuscript first. And now, we are applying for English editing service.

We thank you for the invaluable comments and helpful suggestions that contributed to revision of our manuscript.

Round 2

Reviewer 2 Report

The manuscript in this version is suitable for publication